# ADAPTIVE CORRECTION MECHANISM FOR ENSURING CONSERVATION LAWS IN NEURAL OPERATORS

## ABSTRACT

Physical laws, such as the conversation of mass and momentum, are fundamental principles in many physical systems. Neural operators have achieved promising performance in learning the solutions to those systems, but they often fail to ensure conversation. Existing methods typically enforce conservation via hand-crafted post-processing or architectural constraints, leading to limited model flexibility and adaptability. In this work, we propose a novel adaptive correction approach to ensure the conservation of fundamental quantities for neural operator outputs. Our method introduces a lightweight learnable operator to adaptively enforce the target conservation law during training. This mechanism allows the model to flexibly and adaptively correct its outputs while guaranteeing conservation. We provide a theoretical guarantee showing that neural operators with our correction method can potentially achieve lower reconstruction loss than their conservation-constrained counterparts. Our method is evaluated across multiple neural operator architectures and representative PDEs. Extensive experiments show that incorporating our correction method into baseline models significantly improves both accuracy and stability. In addition, the experimental results demonstrate that our approach consistently achieves superior performance over widely used conservation-enforcement techniques on various PDE benchmarks.

## 1 INTRODUCTION

In recent literature, there are a number of neural network models that aim to solve partial differential equations (PDEs), offering a flexible alternative to traditional numerical solvers (Azizzadenesheli et al., 2024; Farea et al., 2024). Traditional numerical methods rely on fine discretizations and handcrafted meshes, which are computationally expensive, particularly in high-dimensional problems or those involving complex geometries and heterogeneous coefficients (Hughes, 2003; Johnson, 2009). In contrast, neural networks can learn solution operators directly from data, enabling mesh-free approximations and rapid inference across diverse inputs and resolutions (Faroughi et al., 2024; Liu et al., 2025; Lu et al., 2021). The data-driven paradigm has led to the development of a variety of neural network architectures and hybrid models (Hafiz et al., 2024). These approaches aim to capture both the physical structure and the dynamic behavior of PDE systems while enhancing scalability and generalization. Neural operators have been instantiated through various architectures, including convolutional (Raonic et al., 2024; Ronneberger et al., 2015), Fourier-based (Li et al., 2021a), transformer-based (Cao, 2021), and graph-based networks (Sharma et al., 2024).

Despite their empirical success, standard data-driven neural operators lack inherent mechanisms to enforce physical conservation laws such as mass, momentum, or energy conservation. Conservation has long been a central research topic in traditional numerical methods, and many classical schemes are specifically designed to satisfy such laws. In contrast, neural operators primarily focus on data approximation while often overlooking the underlying physical principles. This deficiency can lead to non-physical solutions, particularly in domains such as fluid dynamics, plasma physics, and wave propagation, where conservation is essential for physical fidelity and long-term stability. Violations of conservation laws not only reduce solution accuracy but also cause error accumulation over time, ultimately undermining predictive reliability in long-time simulations.

**Limitations in recent conservation methods** To incorporate conservation laws into neural networks, previous work proposed a variety of strategies, which can be broadly categorized into soft and

hard constraint methods. Soft-constraint approaches typically add loss terms to penalize violations of conservation laws, for example, by enforcing mass or energy conservation through integral loss functions (Li et al., 2021b; Wang et al., 2021; Wu et al., 2022). These methods can encourage physical consistency but cannot guarantee exact conservation, especially in long-term simulations. Ideally, constraint-enforcing methods should preserve physical laws while also enhancing model performance. In practice, however, soft-constraint approaches often improve conservation only at the expense of numerical accuracy and stability. Hard-constraint methods enforce conservation laws exactly, either by incorporating correction steps or post-processing mechanisms within the network pipeline (Cardoso-Bihlo & Bihlo, 2025; Geng et al., 2024), or by modifying the internal architecture to explicitly encode conservation properties (Liu et al., 2023b;a; Richter-Powell et al., 2022). While more rigorous in principle, post-processing-based methods typically rely on fixed, hand-crafted procedures that lack adaptability to diverse inputs. Architecture-based approaches are often restricted to enforcing linear conservation laws and may be incompatible with advanced or modular network architectures.

**Our adaptive correction method** In this work, we introduce an adaptive correction approach to enforce conservation laws in neural operators. Our method incorporates a lightweight learnable operator that ensures the outputs of neural operators strictly satisfy conservation constraints. Importantly, our approach fully preserves the architecture of the original model and does not compromise its expressive power. Our method operates efficiently with various neural operator architectures, including CNN-, Transformer-, and Fourier-based models. Comprehensive experiments across a variety of neural operators and PDEs demonstrate that our approach can enforce exact conservation in neural operators while also improving their predictive accuracy and stability.

## 2 RELATED WORK

Enforcing conservation laws in neural network-based PDE solvers has been an active research direction. One common approach is to augment the original data loss with an additional conservation loss term to enforce conservation constraints:

$$||u(\boldsymbol{x},t) - u_{gt}(\boldsymbol{x},t)||_{L^2} + \lambda ||\mathcal{G}(u)||_{L^2}, \tag{1}$$

where $u_{gt}(\boldsymbol{x},t) : \Omega \times [0,T] \to \mathbb{R}^N$ and $u(\boldsymbol{x},t) : \Omega \times [0,T] \to \mathbb{R}^N$ denote the ground-truth solution and the neural network approximation, respectively, and $\mathcal{G}(u) = 0$ encodes the conservation law to be enforced. This technique is widely used to encourage approximate satisfaction of conservation laws (Chen & Qiao, 2025; De Ryck et al., 2024; Saharia et al., 2024; Wu et al., 2022). While such loss functions can improve the conservation behavior of models, they require careful tuning of the penalty weight $\lambda$ and cannot guarantee strict conservation. For physics-informed neural operators (Li et al., 2021b; Wang et al., 2021), their performance is highly sensitive to the choice of $\lambda$: even small variations can lead to significant degradation, as demonstrated in Section 4.3. This sensitivity substantially increases the difficulty of model tuning.

Exact preservation in neural networks for dynamical systems has recently been studied using various mathematical techniques. In Müller (2023), Noether's theorem is used to enforce Lie symmetries within neural networks. For Lagrangian systems, this approach guarantees the exact conservation of the associated first integrals. However, a key limitation of this method is that it applies only to Lagrangian systems, as non-Lagrangian systems fall outside the scope of Noether's theorem. A more general method was proposed in (Cardoso-Bihlo & Bihlo, 2025), which incorporates the projection method into the training process. This method corrects the output at each step by solving a constrained optimization problem to enforce conservation laws, offering a more flexible approach that can be applied to a broader range of systems. More precisely, this method corrects the output by solving the following minimization problem:

$$\min_{u} ||u(\boldsymbol{x},t) - \hat{u}(\boldsymbol{x},t)||_2$$
$$s.t. \quad \mathcal{G}(u) = 0. \tag{2}$$

Here, $\hat{u}$ is the output of a given neural network and $\mathcal{G}(u) = 0$ encodes the conservation law. While this method can achieve strict conservation, it incurs high computational costs, as each step requires solving a minimization problem. Moreover, its convergence is often difficult to guarantee, making it less practical for large-scale problems.

Another line of work introduces conservation constraints via the divergence of skew-symmetric matrix-valued functions, effectively enforcing linear conservation laws, such as momentum conservation (Liu et al., 2023a; Richter-Powell et al., 2022). However, these methods are inherently limited to linear quantities and cannot generalize to nonlinear conservation laws such as norm or energy conservation.

Recently, a simple but effective postprocessing method was proposed to ensure mass conservation in neural operator outputs (Geng et al., 2024). This constant adjustment method works by computing the mass discrepancy between the initial condition and the prediction and adds it to the latter. While computationally efficient and exact for mass conservation, it is inherently limited to linear conserved quantities and does not extend to nonlinear constraints. However, both projection-based and adjustment-based corrections are static, non-learnable postprocessing steps that lack adaptability to diverse input conditions. In this work, we propose an adaptive correction framework that overcomes these limitations by introducing a learnable operator to adjust neural operator outputs in a data-driven and input-dependent manner. Our method enforces conservation laws exactly while preserving and even enhancing predictive accuracy, thereby providing a more flexible, scalable, and physically consistent alternative to conventional correction techniques.

## 3 THE ADAPTIVE CORRECTION METHOD

To ensure that neural operators respect fundamental conservation laws in physics, we introduce an adaptive correction method, which can handle two types of conservation laws in closed physical systems: linear and quadratic.

### 3.1 CORRECTION FOR LINEAR CONSERVATION LAWS

Linear conservation laws, such as mass and momentum conservation, are characterized by the invariance of linear integral quantities over time. In mathematical terms, these laws require the following condition.

$$\frac{d}{dt} \int_{\Omega} u(\boldsymbol{x}, t) \, d\boldsymbol{x} = 0. \tag{3}$$

Here, $u(\boldsymbol{x}, t)$ can represent a single physical quantity or the product of two quantities, for example, density for mass conservation, or density multiplied by velocity for momentum conservation. This equation implies the $\int_{\Omega} u(\boldsymbol{x}, t) \, d\boldsymbol{x}$ is constant over time, which we denote as $m_0$.

For simplicity, we consider the one-dimensional case. We first divide the spatial domain into $N$ equal small regions. Denote the $i$-th region as $\Delta x_i$. Suppose $U_i(t)$ is a discrete approximation of $\int_{\Delta x_i} u(\boldsymbol{x}, t) \, d\boldsymbol{x}$. Then discrete formulation of equation 3 can be characterized as follows.

$$\sum_{i=1}^{N} U_i = m_0. \tag{4}$$

To ensure that equation 4 is satisfied for neural network output, we first propose a series of local correction operators as

$$\{\mathcal{L}_i\}_{i=1}^{N}, \quad \mathcal{L}_i : (m_0, \boldsymbol{U}) \to \boldsymbol{U}_{\text{new}},$$

where $\boldsymbol{U}$ is the discrete solution given by the neural operator and $\boldsymbol{U} = (U_1, U_2, \cdots, U_N)$. These operators are designed to map the original output $\boldsymbol{U}$ to a new output $\boldsymbol{U}_{\text{new}}$ that exactly satisfies equation 4. They are defined as follows

$$[\mathcal{L}_i(m_0, \boldsymbol{U})]_k = \begin{cases} U_k, & \text{if } k \neq i, \\ m_0 - \sum_{k \neq i} U_k & \text{if } k = i. \end{cases} \tag{5}$$

Each operator $\mathcal{L}_i$ alters only the $i$-th entry of $\boldsymbol{U}$ to guarantee conservation, with all other entries unchanged.

In particular, when $\boldsymbol{U}$ represents multiple quantities, as in the case of momentum conservation laws, $\boldsymbol{U} = \boldsymbol{\rho} \cdot \boldsymbol{v}$, there exists an infinite number of solution pairs $(\boldsymbol{\rho}, \boldsymbol{v})$ to generate $\boldsymbol{U}_{\text{new}}$. One valid solution can be constructed, for example, by modifying only a single quantity at the $i$-th entry while keeping all other quantities fixed.

Relying solely on one local correction operator sacrifices flexibility at that point, as the value at that position is merely determined by the others. To overcome this, we combine all local correction operators to obtain a global correction operator. Specifically, we define the global mass-conserving operator as

$$\mathcal{L}(m_0, \boldsymbol{U}) = \sum_{i=1}^{N} \alpha_i \mathcal{L}_i(m_0, \boldsymbol{U}), \tag{6}$$

where the coefficients $\alpha_i$ satisfy the constraint:

$$\sum_{i=1}^{N} \alpha_i = 1. \tag{7}$$

With this constraint, the updated output $\boldsymbol{U}_{\text{new}}$ obtained by $\mathcal{L}(m_0, \boldsymbol{U})$ also satisfies equation 4. By parameterizing $\alpha_i$ with a softmax function, the constraint is naturally satisfied while simultaneously enabling our global correction operator to be learnable.

Moreover, by substituting equation 5 into equation 6, we can simplify the expression and obtain

$$\mathcal{L}(m_0, \boldsymbol{U}) = \boldsymbol{U} + (m_0 - M(\boldsymbol{U}))\boldsymbol{A} \tag{8}$$

where $M(\boldsymbol{U}) = \sum_{i=1}^{N} U_i$ and $\boldsymbol{A}(k) = \alpha_k$. This formulation reveals that the global correction operator admits a straightforward implementation using the original output $\boldsymbol{U}$ and a learnable vector $\boldsymbol{A}$.

## 3.2 CORRECTION FOR QUADRATIC CONSERVATION LAWS

Quadratic conservation laws, such as energy or norm conservation, require invariance of quadratic quantities. A canonical example is the conservation of the squared norm:

$$\frac{d}{dt} \int_{\Omega} |u(\boldsymbol{x}, t)|^2 \, d\boldsymbol{x} = 0. \tag{9}$$

This equation implies that the quantity $\int_{\Omega} |u(\boldsymbol{x}, t)|^2 \, d\boldsymbol{x}$ remains constant. Let this constant be denoted as $c_0$. The corresponding 1D discrete form is given by

$$\sum_{i=1}^{N} U_i^2 = c_0. \tag{10}$$

Unlike the linear case, applying a series of conserved operators does not guarantee a conserved output for quadratic laws. Inspired by the form of equation 8 in the linear case, we introduce a quadratic correction operator $\mathcal{L}^q : (c_0, \boldsymbol{U}) \to \boldsymbol{U}_{\text{new}}$ by assuming $\boldsymbol{U}_{\text{new}}$ is a combination of the neural operator output $\boldsymbol{U}$ with a learnable vector $\boldsymbol{A}$:

$$\boldsymbol{U}_{\text{new}} = \lambda_1 \boldsymbol{U} + \lambda_2 \boldsymbol{A}. \tag{11}$$

To ensure that equation 10 is satisfied, the following condition must be met:

$$\sum_{i}^{N} (\lambda_1 U_i + \lambda_2 A_i)^2 = c_0. \tag{12}$$

Define the following quantities

$$S_{U^2} = \sum_{i}^{N} U_i^2, \quad S_{A^2} = \sum_{i}^{N} A_i^2, \quad S_{UA} = \sum_{i}^{N} U_i A_i. \tag{13}$$

equation 12 then becomes

$$\lambda_1^2 S_{U^2} + 2\lambda_1 \lambda_2 S_{UA} + \lambda_2^2 S_{A^2} = c_0. \tag{14}$$

A real-valued solution for $\lambda_2$ exists if and only if

$$(2\lambda_1 S_{UA})^2 - 4S_{A^2}(\lambda_1^2 S_{U^2} - c_0) \geq 0. \tag{15}$$

Note that when $\lambda_1^2 S_{U^2} - c_0 \leq 0$, equation 15 always holds. In this case, equation 14 admits a real solution for $\lambda_2$, which can be obtained in closed form:

$$\lambda_2 = \frac{-\lambda_1 S_{UA} \pm \sqrt{(\lambda_1 S_{UA})^2 - S_{A^2}(\lambda_1^2 S_{U^2} - c_0)}}{S_{A^2}}. \tag{16}$$

To simplify the solution and ensure guaranteed feasibility, we assume $\lambda_1^2 S_{U^2} - c_0 = 0$, then we have,

$$\lambda_1 = \pm\sqrt{\frac{c_0}{S_{U^2}}}, \quad \lambda_2 = \mp\frac{2S_{UA}}{S_{A^2}}$$

Thus, we define the following operator $\mathcal{L}^q : (c_0, \boldsymbol{U}) \to \boldsymbol{U}_{\text{new}}$ for quadratic conservation laws.

$$\mathcal{L}^q(c_0, \boldsymbol{U}) = \sqrt{\frac{c_0}{S_{U^2} + \epsilon}}\boldsymbol{U} - \frac{2S_{UA}}{S_{A^2} + \epsilon}\boldsymbol{A}, \tag{17}$$

where $\epsilon > 0$ is a small constant added for numerical stability, preventing division by zero when $S_{U^2}$ or $S_{A^2}$ approaches zero.

**Theorem 1** *Define the following loss functions:*

$$L_1(u, u_{gt}) = \|u - u_{gt}\|,$$

$$L_2(u) = \begin{cases} \infty, & \mathcal{G}(u) \neq 0, \\ 0, & \mathcal{G}(u) = 0. \end{cases} \tag{18}$$

*Let $\mathcal{N}_F^\theta$ be the original neural operator, and $\mathcal{N}_A^\theta$ be the neural operator with our proposed adaptive correction. Define:*

$$\mathcal{N}_F^* = \arg\min_{\mathcal{N}_F^\theta} L_1(\mathcal{N}_F^\theta(u_0), u_{gt}) + L_2(\mathcal{N}_F^\theta(u_0)),$$

$$\mathcal{N}_A^* = \arg\min_{\mathcal{N}_A^\theta} L_1(\mathcal{N}_A^\theta(u_0), u_{gt}).$$

*We have*

$$L_1(\mathcal{N}_A^*(u_0), u_{gt}) \leq L_1(\mathcal{N}_F^*(u_0), u_{gt}). \tag{19}$$

See Appendix C for the detailed proof.

**Remark 1** *Directly enforcing conservation by optimizing $\mathcal{L}_1 + \lambda\|\mathcal{G}(u)\|_{L^2}$ with large $\lambda$ often leads to unstable training. Theorem 1 suggests that, in the extreme case where $\lambda = \infty$, this constrained optimization can be effectively replaced by training a neural operator with our adaptive correction.*

### 3.3 SUMMARY AND IMPLEMENTATION NOTES

Together, the equation 8 and equation 17 define correction operators for linear and quadratic conservation laws respectively. In both cases, learnable coefficients $\boldsymbol{A}$ are introduced to endow the correction mechanism with adaptability and learning capability. In practice, the learnable coefficients $\boldsymbol{A}$ can be implemented as a set of trainable parameters or generated dynamically by a lightweight neural network such as a convolutional layer or a multilayer perceptron (MLP), conditioned on the type of the neural operator. For CNN-based neural operator like UNet, we choose convolutional layers to generate $\boldsymbol{A}$, while in the case of the Fourier Neural Operator (FNO) (Li et al., 2021a), which is resolution-invariant, we adopt an entry-wise MLP to generate $\boldsymbol{A}$ based on the output, which ensures that the correction mechanism inherits the resolution-invariance property of the FNO, thereby preserving their key advantages.

# 4 EXPERIMENTS

We conduct extensive experiments to evaluate the effectiveness of the proposed adaptive correction method in enforcing mass and norm conservation across a variety of neural operator architectures and benchmark PDEs. Specifically, we evaluate our method on three representative neural operators: UNet (Ronneberger et al., 2015), Galerkin Transformer Neural Operator (GTNO) (Cao, 2021), and Fourier Neural Operator (FNO) Li et al. (2021a). Our implementation of the GTNO and the FNO builds upon the publicly available codebases at `https://github.com/scaomath/galerkin-transformer` and `https://github.com/neuraloperator/neuraloperator`, respectively. More implementation details are provided in Appendix A. The learnable coefficients $\boldsymbol{A}$ are parameterized by a single convolutional layer for UNet and GTNO, and by a lightweight MLP with three hidden layers for FNO, and this setup is used consistently across all experiments. The results demonstrate that our method not only strictly preserves the targeted conservation laws but also consistently improves predictive accuracy over the original models. We further compare our method with loss-based correction and projection-based correction on FNO to highlight its advantages.

## 4.1 BENCHMARK PDEs

We select three mass-conserving and three norm-conserving equations to evaluate the effectiveness of the proposed adaptive correction method on each type of conservation law. The selected equations are listed below and the data generation details are provided in Appendix B.

**Mass Conservation Equations**

- **Transport Equation (TE):**

$$u_t + \nabla \cdot (u\boldsymbol{v}) = 0, \quad \boldsymbol{x} \in \Omega, \quad t > 0, \tag{20}$$

where $u = u(\boldsymbol{x}, t)$ represents the scalar field, $\boldsymbol{v} = \boldsymbol{v}(\boldsymbol{x}, t)$ is the velocity field, $\boldsymbol{x} \in \mathbb{R}^d$ denotes the spatial coordinates, and $t$ is time. This equation describes the advection of a scalar field and and the mass conservation should be satisfied when the system is closed or the boundary condition is periodic. We test the 2D transport equation with $\boldsymbol{v}(\boldsymbol{x}, t) \equiv (1, 1)$ on $\Omega = [0, 1]^2$ with the periodic boundary condition. The operator we aim to learn is the mapping between the initial condition $u(\boldsymbol{x}, 0)$ to $u(\boldsymbol{x}, \Delta t)$ with $\Delta t = 0.05$.

- **Conservative Allen-Cahn Equation (CAC):**

$$u_t = \nabla \cdot (\epsilon \nabla u) + u - u^3 - \frac{1}{|\Omega|} \int_\Omega u - u^3 d\boldsymbol{x}, \quad \boldsymbol{x} \in \Omega, \quad t > 0, \tag{21}$$

where $u = u(\boldsymbol{x}, t)$ is the scalar field, $\epsilon > 0$ is a small constant related to interface thickness. This phase-field model is used in material science, and the conservation of mass is critical for accurate simulations. We test the 2D Allen-Cahn Conservative Equation with $\epsilon = 0.01$ on $\Omega = [0, 1]^2$ with the periodic boundary condition. The mapping to be learned is $u(\boldsymbol{x}, 0) \to u(\boldsymbol{x}, \Delta t)$ with $\Delta t = 0.5$.

- **Shallow Water Equations (SWE):**

$$\begin{cases} h_t + \nabla \cdot (h\boldsymbol{u}) = 0, \\ (h\boldsymbol{u})_t + \nabla \cdot \left(h\boldsymbol{u} \otimes \boldsymbol{u} + \frac{1}{2}gh^2\boldsymbol{I}\right) = 0, \end{cases} \quad \boldsymbol{x} \in \Omega, \quad t > 0, \tag{22}$$

Here, $h = h(\boldsymbol{x}, t)$ is the fluid height, $\boldsymbol{u} = \boldsymbol{u}(\boldsymbol{x}, t)$ is the velocity field, $g$ is the gravitational constant, and $\boldsymbol{I}$ denotes the identity matrix. This system models the dynamics of incompressible, inviscid shallow water flow. The first equation enforces mass conservation, which holds under periodic or no-flux boundary conditions. In our experiments, we consider the 2D shallow water equations with periodic boundary conditions. The dataset is obtained from the PDEBench dataset Takamoto et al. (2022). We aim to learn the mapping $h(\boldsymbol{x}, 0) \to h(\boldsymbol{x}, \Delta t)$ with $\Delta t = 0.01$.

| Conservation | Equation | UNet | | GTNO | | FNO | |
|---|---|---|---|---|---|---|---|
| | | original | ours | original | ours | original | ours |
| Mass | TE | $1.08 \pm 0.14$e-1 | $\mathbf{0.83 \pm 0.09}$e-1 | $9.11 \pm 0.76$e-2 | $\mathbf{8.15 \pm 0.87}$e-2 | $8.29 \pm 0.12$e-2 | $\mathbf{8.04 \pm 0.11}$e-2 |
| | CAC | $1.48 \pm 0.08$e-2 | $\mathbf{1.42 \pm 0.14}$e-2 | $4.95 \pm 0.17$e-2 | $\mathbf{4.47 \pm 0.18}$e-2 | $2.01 \pm 0.26$e-2 | $\mathbf{1.65 \pm 0.19}$e-2 |
| | SWE | $2.78 \pm 0.29$e-3 | $\mathbf{1.40 \pm 0.08}$e-3 | $1.55 \pm 0.14$e-2 | $\mathbf{0.78 \pm 0.04}$e-2 | $2.57 \pm 0.09$e-3 | $\mathbf{2.32 \pm 0.14}$e-3 |
| Norm | TE | $1.08 \pm 0.14$e-1 | $\mathbf{0.82 \pm 0.05}$e-1 | $9.11 \pm 0.76$e-2 | $\mathbf{8.21 \pm 0.89}$e-2 | $8.29 \pm 0.12$e-2 | $\mathbf{8.01 \pm 0.16}$e-2 |
| | LSE | $1.84 \pm 0.07$e-1 | $\mathbf{1.10 \pm 0.17}$e-1 | $1.66 \pm 0.21$e-3 | $\mathbf{0.80 \pm 0.10}$e-3 | $3.77 \pm 0.28$e-3 | $\mathbf{3.22 \pm 0.09}$e-3 |
| | NSE | $2.40 \pm 0.03$e-1 | $\mathbf{2.33 \pm 0.12}$e-1 | $1.77 \pm 0.10$e-2 | $\mathbf{1.43 \pm 0.04}$e-2 | $3.82 \pm 1.06$e-2 | $\mathbf{3.02 \pm 0.51}$e-2 |

Table 1: Prediction error on test dataset for the original neural operators and their counterparts with the proposed adaptive correction method.

**Norm Conservation Equations**

- **Transport Equation (TE):** As in the setting of the mass conservation case, the $L^2$ norm of $u$ is also conserved over time.

- **Schrödinger Equation:**

$$\text{Linear:} \quad i\psi_t + \frac{1}{2}\Delta\psi + V(\boldsymbol{x})\psi = 0, \quad \boldsymbol{x} \in \Omega, \quad t > 0, \qquad \text{(LSE)}$$

$$\text{Nonlinear:} \quad i\psi_t + \frac{1}{2}\Delta\psi + \lambda||\psi||^2\psi = 0, \quad \boldsymbol{x} \in \Omega, \quad t > 0, \qquad \text{(NSE)}$$

(23)

where $\psi = \psi(\boldsymbol{x}, t)$ is a complex-valued wave function, $i$ is the unit imaginary number and $\Delta$ denotes the Laplacian operator. A key property of both the linear and nonlinear Schrödinger equations is the conservation of the $L^2$ norm $\int_\Omega |\psi(\boldsymbol{x}, t)|^2 d\boldsymbol{x}$, which is fundamental in quantum mechanics. In our experiments, $\Omega = [0, 1]$ and we use $V(\boldsymbol{x}) \equiv 1$ for the linear case and set $\lambda = 1$ for the nonlinear case, with periodic boundary conditions in both. The mapping we aim to learn is $\psi(\boldsymbol{x}, 0) \to \psi(\boldsymbol{x}, \Delta t)$ with $\Delta t = 0.025$.

## 4.2 ACCURACY TESTS FOR ADAPTIVE CORRECTION

We evaluate prediction accuracy for all baseline models, with results reported in Table 1. All neural operators equipped with the proposed adaptive correction outperform their original counterparts on all test benchmarks. In addition to prediction accuracy, our method also improves the preservation of inherent physical properties. Figure 1 shows the evolution of the $L^2$ norm of $\psi$, which describes mass density or probabilistic density, for both FNO and FNO with our adaptive correction. The standard FNO deviates from the ground truth and exhibits error growth over time, whereas the FNO with adaptive correction remains closely aligned with the ground truth even after multiple prediction steps. Figure 2 illustrates how prediction errors evolve over time: while the standard FNO quickly diverges from the ground truth, the corrected FNO remains closely aligned, demonstrating that our method enhances the stability of neural operators in long-term prediction.

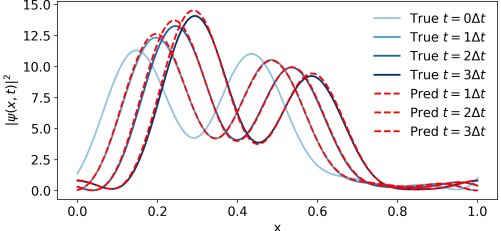 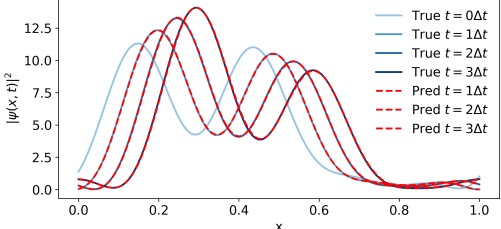

Figure 1: Solution dynamics of the linear Schrödinger equation obtained with the baseline FNO and our proposed method over time, starting from $t = 0$ (solid light blue line). $\Delta t$ denotes the prediction time interval. Left: FNO. Right: FNO with our method.

## 4.3 ADAPTIVE CORRECTION VS. OTHER METHODS

In this subsection, we compare the proposed adaptive correction method with the loss-based method and the projection method on the FNO model. The prediction error is calculated as the relative

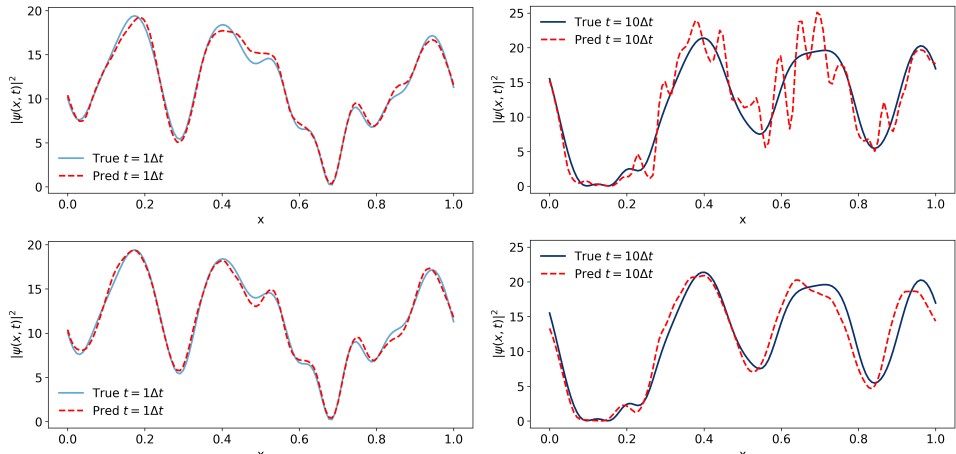

Figure 2: Predictions for the nonlinear Schrödinger equation. Top: FNO. Bottom: FNO with our method. Left: one-step prediction at $t = \Delta t$ (both methods show small error). Right: ten-step prediction at $t = 10\Delta t$ (FNO's error is amplified; our method remains stable). $\Delta t$ denotes the prediction time interval.

| Conservation Laws | Equation | FNO | Loss | Projection | Ours |
|---|---|---|---|---|---|
| Mass | TE | $8.29 \pm 0.12$e-2 | $8.16 \pm 0.11$e-2 | $8.14 \pm 0.14$e-2 | $\mathbf{8.04 \pm 0.11}$e-2 |
| | CAC | $2.01 \pm 0.26$e-2 | $2.24 \pm 0.34$e-2 | $99.7 \pm 0.47$e-2 | $\mathbf{1.65 \pm 0.19}$e-2 |
| | SWE | $2.57 \pm 0.09$e-3 | $2.82 \pm 0.14$e-3 | $3.01 \pm 0.34$e-3 | $\mathbf{2.32 \pm 0.14}$e-3 |
| Norm | TE | $8.29 \pm 0.12$e-2 | $8.17 \pm 0.24$e-2 | $8.34 \pm 0.24$e-2 | $\mathbf{8.01 \pm 0.16}$e-2 |
| | LSE | $3.77 \pm 0.28$e-3 | $4.08 \pm 0.87$e-3 | $3.94 \pm 0.98$e-3 | $\mathbf{3.22 \pm 0.09}$e-3 |
| | NSE | $3.82 \pm 1.06$e-2 | $3.75 \pm 0.32$e-2 | $3.52 \pm 0.35$e-2 | $\mathbf{3.02 \pm 0.51}$e-2 |

Table 2: Prediction error on test dataset for the original FNO and FNO with conservation methods.

$L^2$ error between the predicted and true solutions, while the conservation error is calculated as the discrepancy between the sum of the conservative quantity at all grid points. All results are reported in Table 2 and Table 3, respectively.

**Adaptive Correction vs. Loss-based method** The loss-based method typically augments the training objective by incorporating an additional term that penalizes violations of conservation laws. Specifically, we implement this approach by training the FNO using the loss function defined in Equation 1. We first evaluate the loss-based approach on the transport equation under various choices of $\lambda$. The results for mass conservation and norm conservation are presented in Table 4. The findings indicate that careful tuning of $\lambda$ is crucial, as values that are too large can adversely affect performance. Additionally, the results do not vary smoothly with changes in $\lambda$ and slight perturbations in $\lambda$ can cause abrupt changes in performance, as demonstrated by the significant fluctuation observed at $\lambda = 10^{-3}$ in the norm conservation case. Based on the empirically best $\lambda$ values for the transport equation (e.g., $10^{-3}$ for mass conservation and $10^{-4}$ for norm conservation), we evaluate the same loss-based method on other PDEs using these fixed $\lambda$ values. As shown in Table 2 and Table 3, while empirically selected $\lambda$ values yield acceptable performance for the transport equation, they do not generalize well to other equations. This underscores a key limitation of loss-based methods: their

| Conservation Laws | Equation | FNO | Loss | Projection | Ours |
|---|---|---|---|---|---|
| Mass | TE | $6.42 \pm 1.2$ | $5.27 \pm 1.6$ | $\mathbf{0.00 \pm 0.0}$ | $\mathbf{0.00 \pm 0.0}$ |
| | CAC | $46.7 \pm 7.4$ | $41.7 \pm 5.2$ | $\mathbf{0.00 \pm 0.0}$ | $\mathbf{0.00 \pm 0.0}$ |
| | SWE | $13.3 \pm 1.4$ | $9.72 \pm 0.9$ | $\mathbf{0.00 \pm 0.0}$ | $\mathbf{0.00 \pm 0.0}$ |
| Norm | TE | $31.6 \pm 5.4$ | $26.2 \pm 5.8$ | $\mathbf{0.00 \pm 0.0}$ | $\mathbf{0.00 \pm 0.0}$ |
| | LSE | $2.55 \pm 0.4$ | $2.27 \pm 0.5$ | $\mathbf{0.00 \pm 0.0}$ | $\mathbf{0.00 \pm 0.0}$ |
| | NSE | $13.5 \pm 6.2$ | $11.2 \pm 4.7$ | $\mathbf{0.00 \pm 0.0}$ | $\mathbf{0.00 \pm 0.0}$ |

Table 3: Conservation error on test dataset for the original FNO and FNO with conservation methods.

| $\lambda$ | 0 | 1e-4 | 1e-3 | 1e-2 |
|---|---|---|---|---|
| Mass conservation | $8.29 \pm 0.12$ | $8.25 \pm 0.15$ | $\mathbf{8.16 \pm 0.11}$ | $9.29 \pm 0.12$ |
| $\lambda$ | 0 | 1e-5 | 1e-4 | 1e-3 |
| Norm conservation | $8.29 \pm 0.12$ | $8.35 \pm 0.21$ | $\mathbf{8.17 \pm 0.16}$ | $90.1 \pm 0.32$ |

Table 4: Prediction error (%) for the FNO with different conservation loss and different $\lambda$.

performance heavily depends on carefully tuned hyperparameters, which may not be transferable across problem settings.

In contrast, our adaptive correction method does not require manual parameter tuning and consistently outperforms both the original FNO and the loss-based method across all evaluated equations. Notably, while the loss-based method does lead to a slight reduction in conservation error, our method drives the conservation error down to machine precision, which achieves exact satisfaction of the conservation law in the output.

**Adaptive Correction vs. Projection method** The projection methods are implemented by solving equation 2 both in training and in prediction. As shown in Table 2 and Table 3, while the projection method enforces exact conservation, it does not reduce the prediction error and, in the case of the conservative Allen–Cahn equation, significantly increases it. In contrast, our adaptive correction method not only enforces conservation laws effectively but also maintains or improves predictive accuracy compared to both baselines. Notably, the conservation error with our method is consistently at machine precision, highlighting its ability to enforce physical constraints without sacrificing accuracy.

## 4.4 Ablation Study

We conducted an ablation study to verify that the observed performance gains stem from the targeted enforcement of conservation laws rather than the mere addition of learnable parameters. To this end, we compared our approach with a baseline that appends the same MLP to the output of the original FNO architecture without any conservation-driven correction. The results, summarized in Table 5, show that our method consistently outperforms the baseline.

| Conservation Laws | Equation | FNO | FNO$^*$ | Ours |
|---|---|---|---|---|
| Mass Conservation | Transport Equation | $8.29 \pm 0.12$ | $8.26 \pm 0.18$ | $\mathbf{8.04 \pm 0.11}$ |
| | Conservative Allen-Cahn | $2.01 \pm 0.26$ | $2.23 \pm 0.85$ | $\mathbf{1.65 \pm 0.19}$ |
| | Shallow Water Equation | $0.26 \pm 0.01$ | $0.28 \pm 0.01$ | $\mathbf{0.23 \pm 0.01}$ |
| Norm Conservation | Transport Equation | $8.29 \pm 0.12$ | $8.26 \pm 0.18$ | $\mathbf{8.01 \pm 0.16}$ |
| | Linear Schrödinger Equation | $0.38 \pm 0.03$ | $1.61 \pm 0.34$ | $\mathbf{0.32 \pm 0.02}$ |
| | Nonlinear Schrödinger Equation | $3.82 \pm 1.06$ | $4.84 \pm 1.52$ | $\mathbf{3.02 \pm 0.51}$ |

Table 5: Prediction error (%) on test dataset for the original FNO, **FNO$^*$** (FNO with a learnable matrix appended) and FNO with the proposed adaptive correction method.

## 5 Conclusion and Limitations

In this work, we propose an adaptive correction approach that dynamically adjusts the output of neural operators to satisfy conservation laws. We conducted a comprehensive set of experiments on various neural operator architectures and PDEs, demonstrating the effectiveness of our method for mass and norm conservation. The results show that our approach not only exactly enforces the desired conservation laws but also improves the overall accuracy and stability of the prediction. Further comparisons with existing conservation techniques highlight the superiority of our adaptive correction method. At present, our approach focuses on enforcing a single conservation law at a time, and is limited to linear and quadratic forms. Extending the framework to simultaneously enforce multiple conservation laws and to handle higher-order conservation laws represents a promising direction for future work.

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

# Appendix

## A    IMPLEMENTATIONS DETAILS FOR FNO

### A.1    UNET CONFIGURATIONS

The implemented UNet model is a five-level architecture with an initial depth of four encoding layers, followed by a bottleneck, and four decoding layers. The input channels are configurable (default: 3). Starting with 16 feature channels, the number doubles at each encoding step, reaching 256 channels at the bottleneck. Downsampling is performed using max-pooling layers with a kernel size and stride of two, while upsampling uses transposed convolutions with the same parameters. Each convolutional block consists of two convolutional layers with a kernel size of three, circular padding, Tanh activations, and no bias. Skip connections are used to concatenate encoder features with the corresponding decoder layers, enabling the network to integrate hierarchical and spatial details effectively. The output layer uses a $1 \times 1$ convolution to adjust the feature map dimensions to the desired number for outputs. The learning rate for UNet is 1e-4 across all experiments.

### A.2    GTNO CONFIGURATIONS FOR DIFFERENT PDE BENCHMARKS.

Detailed configurations for the GTNO are shown in Table 6 and Table 7.

### A.3    FNO CONFIGURATIONS FOR DIFFERENT PDE BENCHMARKS.

Detailed configurations for the FNO are shown in Table 8 and Table 9.

### A.4    IMPLEMENTATION DETAILS OF THE ADAPTIVE CORRECTION MODULE

For UNet and GTNO, $\boldsymbol{A}$ is parameterized by a convolutional layer with kernel size 3 in all experiments, taking as input the concatenation of the original output and the final feature maps. For FNO, $\boldsymbol{A}$ is parameterized by a lightweight three-layer MLP, whose hidden dimension is set to twice the number of output channels (i.e., only 1 or 2 in our experiments).

### A.5    TRAINING DETAILS

For all experiments, we train the baselines with Adam optimizer and a learning rate decay of 0.5 is applied every 100 epochs. For the training of UNet and GTNO, we train 500 epochs for all equations. For the training of FNO models, we train 100 epochs for the transport equation and 500 epochs for all other equations.

The spatial resolution for 2D equations, including the transport equation, the conservative Allen–Cahn equation, and the shallow water equation, is set to $128 \times 128$. For 1D equations such as linear and nonlinear Schrödinger equations, the data are represented with shape $(2, 128)$ to accommodate complex-valued functions.

We used 2,000 training samples for the transport equation, and 1000 training samples for all other equations. Each experiment was evaluated on an additional 1,000 test samples. All experiments were conducted on a single NVIDIA A100 GPU with 80GB of memory. To ensure the reliability of our results, each experiment is repeated three times using different random seeds.

## B    DATA GENERATION DETAILS FOR PDES

- **Transport Equation:**

$$u_t + \nabla \cdot (u\boldsymbol{v}) = 0, \quad x \in \Omega, \quad t > 0, \tag{24}$$

For the 2D transport equation with constant velocity field $\boldsymbol{v} \equiv (1, 1)$, the analytical solution is given by:

$$u(x, y, t) = u_0(x - t, y - t, 0), \tag{25}$$

Table 6: GTNO Model Hyperparameters for Transport Equation, Conservative Allen-Cahn Equation and Shallow Water Equation

| Parameter | Value |
|-----------|-------|
| learning_ rate | 1e-4 |
| node_feats | 1 |
| pos_dim | 2 |
| n_targets | 1 |
| n_hidden | 128 |
| num_feat_layers | 0 |
| num_encoder_layers | 6 |
| n_head | 4 |
| dim_feedforward | 256 |
| feat_extract_type | null |
| attention_type | galerkin |
| xavier_init | 0.01 |
| diagonal_weight | 0.01 |
| symmetric_init | False |
| layer_norm | False |
| attn_norm | True |
| norm_eps | 0.0000001 |
| batch_norm | False |
| return_attn_weight | False |
| return_latent | False |
| decoder_type | ifft2 |
| spacial_dim | 2 |
| spacial_fc | True |
| upsample_mode | interp |
| downsample_mode | interp |
| freq_dim | 32 |
| boundary_condition | dirichlet |
| num_regressor_layers | 2 |
| fourier_modes | 12 |
| regressor_activation | silu |
| downscaler_activation | relu |
| upscaler_activation | silu |
| last_activation | True |
| dropout | 0.0 |
| downscaler_dropout | 0.05 |
| upscaler_dropout | 0.0 |
| ffn_dropout | 0.05 |
| encoder_dropout | 0.05 |
| decoder_dropout | 0 |
| downscaler_size | 32 |

where the initial condition is defined as:

$$u_0(x, y, t) = A\sin(2\pi k_1 x)\sin(2\pi k_2 y), \tag{26}$$

with the amplitude $A$ sampled uniformly from $[2.5, 3]$ and the wave numbers $k_1, k_2$ randomly selected from $0, 1, 2, 3$.

- **Conservative Allen-Cahn Equation:**

$$u_t = \nabla \cdot (\epsilon \nabla u) + u - u^3 - \frac{1}{|\Omega|}\int_\Omega u - u^3 d\boldsymbol{x}, \quad \boldsymbol{x} \in \Omega, \quad t > 0, \tag{27}$$

We simulate the 2D conservative Allen–Cahn equation using the forward Euler method with a time step size of $10^{-5}$ and $\epsilon = 0.01$. Periodic boundary conditions are applied. The initial condition values are sampled independently and uniformly from the interval $[-1, 1]$ at each spatial point.

Table 7: GTNO Model Hyperparameters for Linear and Nonlinear Schrödinger Equation

| Parameter | Value |
|---|---|
| learning_ rate | 1e-4 |
| node_feats | 2 |
| pos_dim | 1 |
| n_targets | 2 |
| n_hidden | 96 |
| num_feat_layers | 0 |
| num_encoder_layers | 4 |
| n_head | 1 |
| dim_feedforward | 192 |
| feat_extract_type | null |
| attention_type | fourier |
| xavier_init | 0.01 |
| diagonal_weight | 0.01 |
| symmetric_init | False |
| layer_norm | False |
| attn_norm | True |
| norm_eps | 0.0000001 |
| batch_norm | False |
| return_attn_weight | False |
| return_latent | False |
| decoder_type | ifft2 |
| spacial_dim | 2 |
| spacial_fc | True |
| upsample_mode | interp |
| downsample_mode | interp |
| freq_dim | 48 |
| boundary_condition | dirichlet |
| num_regressor_layers | 2 |
| fourier_modes | 16 |
| spacial_dim | 1 |
| spacial_fc | False |
| dropout | 0.0 |
| ffn_dropout | 0.0 |
| encoder_dropout | 0.0 |
| decoder_dropout | 0.0 |

- **Shallow Water Equations:**

$$\begin{cases} h_t + \nabla \cdot (h\boldsymbol{u}) = 0, \\ (h\boldsymbol{u})_t + \nabla \cdot \left(h\boldsymbol{u} \otimes \boldsymbol{u} + \frac{1}{2}gh^2\boldsymbol{I}\right) = 0, \quad x \in \Omega, \quad t > 0, \end{cases} \tag{28}$$

The dataset is obtained from the PDEBench benchmark Takamoto et al. (2022), which simulates a 2D radial dam-break scenario. The initial water height is defined as a circular bump centered in the domain:

$$h(\boldsymbol{x}, t = 0) \begin{cases} 2.0, & r < ||\boldsymbol{x}||, \\ 1.0, & r \geq ||\boldsymbol{x}||, \end{cases} \tag{29}$$

where the radius $r$ is sampled uniformly from the interval $(0.3, 0.7)$.

- **Schrödinger Equations:**

$$\text{Linear:} \quad i\psi_t + \frac{1}{2}\Delta\psi + V(\boldsymbol{x})\psi = 0, \quad \boldsymbol{x} \in \Omega, \quad t > 0,$$

$$\text{Nonlinear:} \quad i\psi_t + \frac{1}{2}\Delta\psi + \lambda||\psi||^2\psi = 0, \quad \boldsymbol{x} \in \Omega, \quad t > 0, \tag{30}$$

Table 8: FNO Model Hyperparameters for Transport Equation, Conservative Allen-Cahn Equation and Shallow Water Equation

| Parameter | Value |
| --- | --- |
| learning_ rate | 1.5e-3 |
| n_modes_height | 48 |
| n_modes_width | 48 |
| in_channels | 3 |
| lifting_channels | 256 |
| hidden_channels | 64 |
| out_channels | 1 |
| projection_channels | 256 |
| n_layers | 4 |
| norm | group_norm |
| skip | linear |
| use_mlp | true |
| factorization | Tucker |
| rank | 1 |

Table 9: FNO Model Hyperparameters for Linear and Nonlinear Schrödinger Equation

| Parameter | Value |
| --- | --- |
| learning_ rate | 1.5e-3 |
| n_modes | 32 |
| in_channels | 4 |
| lifting_channels | 128 |
| hidden_channels | 64 |
| out_channels | 2 |
| projection_channels | 128 |
| n_layers | 4 |
| norm | group_norm |
| skip | linear |
| use_mlp | true |
| factorization | Tucker |
| rank | 1 |

We solve both the linear and nonlinear Schrödinger equations using the Strang splitting method Strang (1968) combined with the Fast Fourier Transform (FFT) Kumar et al. (2019). The initial condition is constructed as:

$$u_0 = \sum_{k=1}^{5} (a_k + b_k i) e^{ikx + \phi_k} \tag{31}$$

where $a_k, b_k \sim \mathcal{N}(0, 1)$ are drawn from a standard normal distribution and $\phi_k \sim \mathcal{U}(0, 2\pi)$ are uniformly sampled phases.

## C   PROOF OF THEOREM 1

Define the following loss functions:

$$L_1(u, u_{gt}) = \|u - u_{gt}\|,$$
$$L_2(u) = \begin{cases} \infty, & \mathcal{G}(u) \neq 0, \\ 0, & \mathcal{G}(u) = 0. \end{cases} \tag{32}$$

Let $\mathcal{N}_F^\theta$ be the original neural operator, and $\mathcal{N}_A^\theta$ be the neural operator with our proposed adaptive correction. Define:

$$\mathcal{N}_F^* = \arg\min_{\mathcal{N}_F^\theta} L_1(\mathcal{N}_F^\theta(u), u_{gt}) + L_2(\mathcal{N}_F^\theta(u)),$$

$$\mathcal{N}_A^* = \arg\min_{\mathcal{N}_A^\theta} L_1(\mathcal{N}_A^\theta(u), u_{gt}).$$

We have

$$L_1(\mathcal{N}_A^*(u), u_{gt}) \le L_1(\mathcal{N}_F^*(u), u_{gt}). \tag{33}$$

**Proof.** Suppose that $\mathcal{N}_F^*$ exists (otherwise $\mathcal{N}_F^* = \emptyset$, equation 33 holds trivially). Then it must satisfy

$$L_2(\mathcal{N}_F^*(u)) = 0,$$

which implies

$$\mathcal{G}(\mathcal{N}_F^*(u)) = 0.$$

*Case 1: Linear conservation law.* Recall that for linear conservation laws,

$$\mathcal{G}(u) = m_0 - M(u).$$

By definition of the linear correction operator $\mathcal{L}$ in equation 8, if $\mathcal{G}(\mathcal{N}_F(u)) = 0$, then $\mathcal{L}$ leaves $\mathcal{N}_F(u)$ unchanged:

$$\mathcal{L}(\mathcal{N}_F^\theta(u)) = \mathcal{N}_F^\theta(u).$$

which implies

$$\mathcal{N}_A^\theta = \mathcal{L}(\mathcal{N}_F^\theta) = \mathcal{N}_F^\theta \text{ if } \mathcal{G}(\mathcal{N}_F^\theta(u)) = 0.$$

Therefore, we have

$$L_1(\mathcal{N}_A^*(u), u_{gt}) = \min_{\mathcal{N}_A^\theta} L_1(\mathcal{N}_A^\theta(u), u_{gt})$$

$$\le \min_{\{\mathcal{N}_A^\theta | \mathcal{G}(\mathcal{N}_F^\theta(u)) = 0\}} L_1(\mathcal{N}_A^\theta(u), u_{gt})$$

$$= \min_{\{\mathcal{N}_F^\theta | \mathcal{G}(\mathcal{N}_F^\theta(u)) = 0\}} L_1(\mathcal{N}_F^\theta(u), u_{gt})$$

$$= L_1(\mathcal{N}_F^*(u), u_{gt}).$$

*Case 2: Quadratic conservation law.* For quadratic conservation laws, if $\mathcal{G}(u) = 0$, then the scaling factor satisfies

$$\frac{c_0}{S_u^2} = 1.$$

Thus, the quadratic correction operator $L^q$ defined in equation 17 maps

$$\mathcal{N}_F^*(u) \to \mathcal{N}_F^*(u) - \frac{2S_{uA}}{S_{A^2} + \epsilon} \boldsymbol{A},$$

Note that for any $u$, $\mathcal{L}$ can also leave $\mathcal{N}_F^\theta(u)$ unchanged:

$$\mathcal{N}_A^\theta(u) = \mathcal{L}(\mathcal{N}_F^\theta(u)) = \mathcal{N}_F^\theta(u) \quad \text{if } \boldsymbol{A} = 0.$$

By similar reasoning as in the linear case, we get

$$L_1(\mathcal{N}_A^*(u), u_{gt}) = \min_{\mathcal{N}_A^\theta} L_1(\mathcal{N}_A^\theta(u), u_{gt})$$

$$\le \min_{\{\mathcal{N}_A^\theta | \boldsymbol{A} = 0\}} L_1(\mathcal{N}_A^\theta(u), u_{gt})$$

$$\le \min_{\{\mathcal{N}_F^\theta | \mathcal{G}(\mathcal{N}_F^\theta(u)) = 0\}} L_1(\mathcal{N}_F^\theta(u), u_{gt})$$

$$= L_1(\mathcal{N}_F^*(u), u_{gt}).$$

This completes the proof.

