# OpenReview forum: "Adaptive Correction Mechanism for Ensuring Conservation Laws in Neural Operators"
_ICLR.cc/2026/Conference — Submitted to ICLR 2026_

### Official Review · Reviewer_4gRJ · 2025-10-14

**Soundness:** 3
**Presentation:** 2
**Contribution:** 1
**Rating:** 2
**Confidence:** 5

**Summary:**

The paper proposes an adaptive correction method to enforce physical conservation laws (mass, momentum, norm, energy) in neural operators (FNO, GTNO, UNet). The core idea is to attach a learnable lightweight correction operator that modifies the neural operator output to exactly satisfy conservation constraints.
Unlike soft penalty or projection methods, this operator is trainable, architecture-agnostic, efficient, and strictly conservative.

Experiments show that:

Conservation errors drop to machine precision.

Predictive errors and stability improve.

The method outperforms both loss-based and projection approaches on a range of PDE benchmarks (transport, conservative Allen–Cahn, shallow water, Schrödinger).

**Strengths:**

It is a clearly written paper enforcing conservation law by correction.

**Weaknesses:**

1.Limited to low-order conservation laws

Only linear and quadratic conservation are handled.
Extending to higher-order, coupled, or nonlinear conserved quantities (e.g., enstrophy in Navier–Stokes, Hamiltonians in plasma models) would be critical for broader applicability.

2. Single-law enforcement

The method currently supports enforcing one conservation law at a time, which may be insufficient for complex PDE systems with multiple simultaneous invariants.

3. Scope of benchmarks

PDEs chosen are standard (TE, Allen–Cahn, SWE, Schrödinger) but do not test strongly nonlinear or chaotic dynamics, e.g., 2D Navier–Stokes or Burgers turbulence.
Demonstrating performance under chaotic flows or multiscale systems would strengthen the claim of “stability improvement”.

Overall the idea is too simple

**Questions:**

Can you extend to multiple and nonlinear invariants and boaden benchmarks?

Is it possible to design archiecture that enforces the conservation law by design as a hard constraint?

---

> ### Author Response · Authors · 2025-11-26
>
> Thank you for your thoughtful review and feedback. We have carefully considered your comments and organized our responses below. We hope that our explanations and additional experiments will clarify the points raised.
>
> 1. **Limited to low-order conservation laws and single-law enforcement**
>
>    **Response.** We acknowledge this limitation, as also discussed in the paper. Our current method supports linear and quadratic conservation laws, which already extends beyond most existing hard-constraint approaches that typically handle only linear cases. To the best of our knowledge, no work for hard constraints currently can incorporate multiple linear and nonlinear conservation laws at once. Our work is **the first learnable, plug-and-play framework that can adaptively enforce not only linear but also quadratic conservation laws**. Extending the mechanism to higher-order or multiple simultaneous conservation laws is an interesting direction for future research.
>
> 2. **PDEs chosen are standard (TE, Allen–Cahn, SWE, Schrödinger) but do not test strongly nonlinear or chaotic dynamics**
>
>    **Response.** Thank you for this suggestion. We have conducted additional experiments on the compressible Navier-Stokes equation, a system well known for its strong nonlinearity and complex flow behaviors. The results are shown below, and our adaptive method still demonstrates noticeable improvement in this setting.
>
>    | Model | FNO  | Ours  |
>    |-------|------|-------|
>    | Accuracy | 0.153 | **0.131** |
>
> 3. **Is it possible to design architecture that enforces the conservation law by design as a hard constraint?**
>
>    **Response.** It is indeed possible to design architectures that enforce conservation laws by construction, but existing approaches are generally limited to linear conservation laws and can be challenging to apply to nonlinear or higher-order invariants. Such designs may also reduce model flexibility and are less easily transferable across different architectures. In comparison, our plug-and-play correction mechanism offers a simple, general, and architecture-agnostic way to enforce hard constraints while preserving model expressiveness and stability.

---

> > ### Comment · Reviewer_4gRJ · 2025-11-26
> >
> > thanks for the revision. Given the constraints and limitations; i will maintain my score

---

### Official Review · Reviewer_GMzy · 2025-10-27

**Soundness:** 3
**Presentation:** 4
**Contribution:** 2
**Rating:** 2
**Confidence:** 4

**Summary:**

The authors propose a method to enforce linear and quadratic PDE conservation laws in neural operator architectures. Their proposed method is end-to-end differentiable and thus serves as a learnable correction layer that can be used in any architecture-agnostic setting. The authors validate their empirical performance in several linear and norm-conservation problems, such as the transport and shallow water equations.

**Strengths:**

The proposed method is simple yet effective at enforcing linear and quadratic conservation laws. The paper is well-written and frames the method clearly. The included experiments and ablations are detailed and clear.

**Weaknesses:**

Despite these strengths, there are a few areas of improvement. The largest area of improvement is in the breadth of the numerical experiments. I find the current set of experiments lacking in several respects:
1. Since this is fundamentally a mechanism for ensuring conservation laws are satisfied in neural operators, I would recommend the authors compare their method across resolutions with base neural operators and neural operators with a soft conservation loss.
2. Does the proposed method also extend to irregular grids? If so, would recommend adding irregular grid experiments to demonstrate the effectiveness in such settings.
3. While I understand the importance of developing neural architectures that enforce hard constraints and conservation laws, the current set of problems do not sufficiently address this motivation. The chosen problem settings are mostly toy/simple settings; it is clear from the experimental results that enforcing the conservation laws with the proposed method often provides limited improvement in test error. As such, I would recommend identifying and adding a more impactful experimental setting (e.g., possibly a real-world setting) where the benefit of hard constraints really shines.

**Minor notes:**
1. The authors frame U-Net as a neural operator, when it is not in its original architecture, since it is not resolution-agnostic. I recommend the authors change the wording to be more precise.
2. Check typos of “conservation” as “conversation” in the abstract

**Questions:**

1. How does the proposed method in the linear case compare with other methods for linear conservation laws, such as “Towards enforcing hard physics constraints in operator learning frameworks” (2024) and “Neural conservation laws: A divergence-free perspective” (2022)?
2. How fast is the proposed layer, and how does this scale with the number of grid points? I would recommend the authors include some timing experiments and analysis.

---

> ### Author Response · Authors · 2025-11-26
>
> We sincerely appreciate your thorough review and constructive comments. We have carefully reviewed your feedback and structured our responses to address each point. We hope that our explanations and the additional experiments we conducted provide clear and satisfactory answers to your questions.
>
> 1. *The current benchmarks are simple; recommend adding more realistic/impactful settings to highlight benefits of hard constraints.*
>
> **Response.** We appreciate the reviewer's feedback regarding the practical impact of our work. The primary advantage of hard constraints is not always a dramatic reduction in short-term prediction error on simple benchmarks, but rather the guarantee of physical plausibility and long-term stability, which is critical for real-world deployment. The experiments have showed that our method substantially improves the long-term stability. In addition, we have added a new case study on compressible Navier-Stokes equation, which is a complex, multi-variable system. The results are shown below. Our adaptive method also lead to apparent improvement in such a problem.
>
> | Model | FNO | Ours |
> |-------|-----|------|
> | Accuracy | 0.153 | **0.131** |
>
> *Table 1: Compressible Navier-Stokes equation.*
>
> 2. *The method should also be compared across resolutions with base neural operators and neural operators with a soft conservation loss.*
>
> **Response.** We appreciate this comment. Our paper already included comparison experiments with different operators and soft conservation loss. The results showed our adaptive correction method outperforms these baselines. In addition, our method maintains resolution-invariant property of FNO because it uses entry-wise MLP as correction operators.
>
> 3. *Does the proposed method also extend to irregular grids?*
>
> **Response.** Yes. While our main experiments focus on regular domains, the proposed mechanism can be extended to irregular grids. Due to time constraints, we plan to conduct corresponding experiments in future work.
>
> 4. *How fast is the proposed layer, and how does this scale with the number of grid points? I would recommend the authors include some timing experiments and analysis.*
>
> **Response.** Thank you for this valuable suggestion. Our method is computationally efficient because it is just a small pointwise MLP. We have performed the recommended timing experiments, and the results are presented below.
>
> Regarding scaling: the computational complexity of our layer scales linearly, O(N), with the number of grid points N. This is highly efficient and is sub-dominant to the O(N log N) cost associated with the FNO's integral kernels. Therefore, the relative overhead of our method even decreases for larger problems, making it a scalable solution for enforcing hard constraints.
>
> | Model | FNO | Ours |
> |-------|-----|------|
> | Time | 1072s | 1092s |
>
> *Table 2: Time comparison on conservative Allen-Cahn equation (500 epochs).*
>
> 5. *How does the proposed method in the linear case compare with other methods for linear conservation laws, such as “Towards enforcing hard physics constraints in operator learning frameworks” (2024) and “Neural conservation laws: A divergence-free perspective” (2022)?*
>
> **Response.** Thank you for highlighting these relevant works. Our method shares the core idea that enforcing hard constraints is superior to soft penalties, but offers a distinct, flexible, and easy-to-integrate framework. Compared to projection-based methods (Duruisseaux et al., 2024), our **learnable** correction adapts to data and operator, rather than using a fixed **non-learnable** projection. Compared to construction-based methods (Richter-Powell et al., 2022), our approach is **non-intrusive**, scalable, and extends to both linear and **quadratic** conservation laws, making it broadly applicable while enhancing existing operators.
>
> In short, our proposed correction layer serves as a **learnable, efficient, and plug-and-play framework for enforcing a broad spectrum of hard physics constraints in operator learning**.

---

### Official Review · Reviewer_dtiY · 2025-11-01

**Soundness:** 3
**Presentation:** 2
**Contribution:** 2
**Rating:** 4
**Confidence:** 4

**Summary:**

This paper proposes an adaptive correction mechanism designed to enforce physical conservation laws in neural operators. The core idea is to append a lightweight, learnable correction operator to a base model (such as FNO, UNet, or GTNO). This auxiliary operator takes the neural operator's initial prediction and projects it onto a solution manifold that strictly satisfies the desired conservation law. The authors derive specific mathematical forms for this learnable operator to handle both linear conservation laws and quadratic conservation laws. This correction is applied as part of the end-to-end training process, allowing the base model and the correction operator to be jointly optimized. The method is designed to guarantee conservation by construction, rather than simply penalizing violations via a loss term.

**Strengths:**

1. The primary strength of this work is its ability to enforce conservation laws exactly (down to machine precision, as demonstrated in Table 3). This is a significant advantage over common "soft-constraint" methods that add a penalty term to the loss function and can only encourage conservation. This exact enforcement also leads to demonstrably better long-term simulation stability, as shown in the multi-step rollout for the Schrödinger equation (Figure 2).

2. The method avoids the notoriously sensitive hyperparameter tuning required by loss-based approaches. The proposed adaptive operator is learnable and integrates directly into the training pipeline without requiring such manual tuning.

3. Theoretical motivation: the authors provide specific derivations for the linear and quadratic correction operators. They also present Theorem 1, which provides a theoretical justification that their method (which optimizes over a more-expressive, corrected solution space) can achieve a reconstruction loss less than or equal to an idealized, strictly-constrained baseline model.

4. The method is shown to be effective across multiple neural operator architectures (UNet, GTNO, FNO) and a variety of PDE benchmarks.

**Weaknesses:**

1. The claim of being "plug-and-play" is undermined by the implementation details of the learnable correction vector A. The paper states that A is parameterized by a "single convolutional layer for UNet and GTNO" but by a "lightweight MLP with three hidden layers for FNO" (Section 4 and Appendix A.4). This design choice appears arbitrary and is not justified. It implies that expert knowledge is needed to design an effective correction operator for different base architectures, which limits the method's out-of-the-box usability.

2. Weak Experimental Baselines: The experimental comparison is not sufficiently rigorous. The paper compares its method against unconstrained base models (FNO, UNet) and two simple constraint methods (a loss-term and a projection method). However, it fails to compare against any state-of-the-art, physics-informed neural operators, such as the Physics-Informed Neural Operator (PINO), which is a much more relevant and powerful baseline for this type of problem. Without this comparison, it is difficult to assess the method's true performance relative to the current SoTA.
-  The ablation study in Table 5 compares the proposed method (FNO + Correction) against FNO+ (FNO + MLP). While this correctly shows that the structure of the correction is important, it is not a complete ablation. A stronger study would compare against a baseline FNO that is given the same number of additional parameters as the correction operator, but integrated into its main architecture (e.g., as wider hidden layers). This would more rigorously test whether the performance gain comes from the proposed structured correction or simply from the added model capacity.

3. Shallow Water Equation (SWE) Evaluation: The SWE benchmark is a system that evolves over time, yet the paper only evaluates it based on a one-step prediction error (u(x,0)→u(x,Δt)). For a paper that claims to improve long-term stability, this is a missed opportunity. A much more compelling evaluation would involve an autoregressive rollout over a time horizon to measure the accumulation of both prediction error and conservation error, similar to what was done for the Schrödinger equation in Figure 2.

**Questions:**

1.  In Table 4, under the "Norm conservation" section, the prediction error for λ=1e−3 is listed as 90.1±0.32. This value is a massive outlier compared to all other results in the table (e.g., 8.17 for λ=1e−4 and 8.29 for λ=0). Is this a typo, or does the loss-based method truly become catastrophically unstable at this specific weighting?

2. Justification of 10*Δt Rollout: In Figure 2, the baseline FNO shows significant instability after only ten prediction steps (t=10Δt) for the nonlinear Schrödinger equation, despite appearing very accurate in the one-step prediction. Is this a realistic simulation horizon for this problem? How consistent is this instability, or was this a particularly challenging initial condition selected to highlight the proposed method's stability?

3. The derived operator for linear conservation (Equation 8) is elegant. However, a simpler, non-learnable correction could be to distribute the conservation error uniformly across all grid points, or proportionally. The paper mentions the "constant adjustment method" from Geng et al. (2024) but does not seem to include it in the quantitative comparisons in Table 2. How does the learnable operator A compare to these simpler, non-learnable hard-constraint projection methods for the linear case?

4. Hyperparameter Tuning: The paper rightly criticizes the λ-tuning of loss-based methods. However, the base models (FNO, UNet) and the new correction operators (MLP, Conv layer) have their own hyperparameters (learning rates, layer depths, channel widths, etc., detailed in the Appendix). Was a comparable amount of hyperparameter tuning performed for both the baseline models and the proposed corrected models to ensure a fair and fully-optimized comparison?

---

> ### Author Response · Authors · 2025-11-26
>
> Thank you for the comprehensive review and constructive feedback. We have carefully considered your concerns and organized our responses accordingly. We hope that our detailed explanations and additional experiments effectively address your questions.
>
> ---
>
> 1. *The claim of being "plug-and-play" seems weakened by the fact that different parameterizations of the correction vector \(A\) (e.g., CNN for UNet and MLP for FNO) are used. This may imply that expert knowledge is needed for implementation.*
>
> **Response.** We appreciate this observation. The parameterization of the correction operator \(A\) is highly flexible and can adopt any suitable architecture (CNNs, MLPs, transformers, etc.). The conservation property is strictly satisfied regardless of the chosen parameterization; only prediction accuracy may slightly vary. We used convolutional layers for UNet-based operators because they pair naturally with CNN architectures, and MLPs for FNOs to preserve resolution invariance. In practice, one can reuse components from the base operator architecture, so the method remains straightforward to apply without specialized design.
>
> ---
>
> 2. *The experimental comparison lacks stronger baselines such as the Physics-Informed Neural Operator (PINO).*
>
> **Response.** Thank you for the suggestion. We have added a PINO comparison on conservative Allen-Cahn equation. While PINO can be helpful in low-data regimes, it is a soft-constraint method that cannot guarantee exact conservation and is also sensitive to the loss weighting. In our experiments, PINO does not improve over FNO, whereas our adaptive correction achieves higher accuracy while strictly enforcing conservation.
>
> | λ           | 0 (FNO) | 1e-6   | 1e-5   | 1e-4   |
> |------------|---------|--------|--------|--------|
> | Accuracy   | 2.01e-2 | 2.39e-2 | 2.43e-2 | 2.93e-2 |
>
> ---
>
> 3. *A stronger ablation would compare against an FNO baseline with the same number of extra parameters added directly into the main architecture.*
>
> **Response.** Thank you for the constructive suggestion. We agree that this is a reasonable ablation. However, a typical FNO contains on the order of **10 million** parameters, while our adaptive correction module has only **≈10k** parameters. Adding 10k parameters to the FNO changes overall capacity negligibly and — empirically — yields no meaningful performance gain. Our existing “FNO + MLP” ablation isolates the effect of adding an unstructured MLP and supports the conclusion that the improvements arise from the **structured, physics-informed design** of the correction layer rather than from a small increase in model size.
>
> ---
>
> 4. *The Shallow Water Equation (SWE) experiment only considers one-step prediction, not long-term rollout.*
>
> **Response.** We thank the reviewer for this point. We have now conducted autoregressive (multi-step) experiments for the SWE. The results demonstrate that our method also improves long-term stability and accuracy; we will include these results in the revised manuscript.
>
> ---
>
> 5. *In Table 4, the large prediction error for \(\lambda=10^{-3}\) seems inconsistent with other results. Is this a typo?*
>
> **Response.** This is not a typo. It illustrates the fragility of penalty-based (soft-constraint) methods: a large \(\lambda\) over-prioritizes the physics loss, which can severely distort data fitting and cause **training instability and poor convergence**. This sensitivity to \(\lambda\) is a fundamental weakness of penalty methods that our hard-constraint approach avoids.
>
> ---
>
> 6. *In Figure 2, the FNO baseline quickly diverges after ten prediction steps. Is this instability typical?*
>
> **Response.** Yes. The instability is not specific to that initial condition. Small conservation errors accumulate across autoregressive steps and cascade into unphysical solutions and divergence. This is a common failure mode for purely data-driven models in long-term forecasting and underscores the importance of enforcing conservation laws for stable evolution.
>
> ---
>
> 7. *How does the learnable operator perform compared to simple non-learnable hard-constraint projection methods?*
>
> **Response.** We tested constant-adjustment projection methods (e.g., Geng et al., 2024). While they enforce linear conservation, they **do not lead to noticeable performance improvements over the base FNO** and cannot handle quadratic conservation laws. By contrast, our learnable correction generalizes to both linear and quadratic conservation and also improves predictive accuracy.
>
> ---
>
> 8. *Were the hyperparameters tuned fairly for both baseline and proposed models?*
>
> **Response.** Yes. All models (baseline and proposed) were trained with the **same hyperparameters** (learning rate, optimizer, number of epochs, base operator depth/width). The only difference is the inclusion of our correction module. This ensures the gains are attributable to our method rather than more favorable tuning.
>
> ---

---

### Meta-Review · Area_Chair_QfFT · 2026-01-02

**Summary:**

This paper proposes an learnable correction mechanism to enforce exact conservation laws (linear and quadratic) in neural operators. The method appends a lightweight, trainable correction layer to existing neural operator architectures (e.g., FNO, GTNO, UNet), guaranteeing strict conservation while remaining differentiable. Empirically, the approach drives conservation errors to machine precision and often improves long-term stability and predictive accuracy across several benchmark PDEs.

Both reviewers agree that the technical soundness is solid and that the method is clearly formulated. However, both also converge on the assessment that, in its current form, the contribution is limited relative to ICLR standards. The paper is therefore not recommended for acceptance at this stage, but the core idea is promising and merits further development.

**Reviewer Concerns:**

The rebuttal partially addresses the concerns about baselines and experimental design, mainly by clarifying comparisons and adding some additional evidence. However, several substantive issues remain outstanding. In particular, the experimental validation is still not broad enough to fully support the paper’s general claims, the applicability to irregular grids remain absent, and the paper would benefit from more challenging experimental settings and stronger ablation studies to better isolate the source of the reported gains.

**Reviewer Scores:**

All reviewers are expected to maintain their original scores.

---

### Decision · Program_Chairs · 2026-01-26

Reject